# Seeing Differently, Acting Similarly: Heterogeneously Observable Imitation Learning

## Abstract

In many real-world imitation learning tasks, the demonstrator and the learner have to act under totally different observation spaces. This situation brings significant obstacles to existing imitation learning approaches, since most of them learn policies under *homogeneous observation spaces*. On the other hand, previous studies under different observation spaces have strong assumptions that these two observation spaces coexist *during the entire learning process*. However, in reality, the observation coexistence will be limited due to the high cost of acquiring expert observations. In this work, we study this challenging problem with limited observation coexistence under heterogeneous observations: *Heterogeneously Observable Imitation Learning* (HOIL). We identify two underlying issues in HOIL, i.e., the dynamics mismatch and the support mismatch, and further propose the *Importance Weighting with REjection* (IWRE) algorithm based on importance-weighting and learning with rejection to solve HOIL problems. Experimental results show that IWRE can successfully solve various HOIL tasks, including the challenging tasks of transforming the vision-based demonstrations to random access memory (RAM)-based policies in the Atari domain, even with limited visual observations.

## 1   Introduction

Imitation Learning (IL) studies how to learn a good policy by imitating the given expert demonstrations [16, 1], and has achieved great success in many domains such as autonomous driving [8], video games [7], and continuous control [19]. In real-world IL applications, the expert and the learner usually have their own observations of the same underlying states from the environment. For example, in Figure 1, an autonomous agent is learning to drive by imitating a human expert. The expert takes her actions mainly based on auditory and visual observations, which are familiar to human beings. However, the learning agent does not necessarily use the same way to observe: it can utilize more machine-capable sensors such as a LiDAR, radar, and bird-eye view (BEV) map to generate its observations  [20]. The key features behind this example are two-fold: First, both the expert and the learner have their *totally different observations* of the *same state* of the environment. Thus they essentially have to choose the same action if acting optimally. Second, the observation space of the expert is often of high cost for the learner to utilize [6, 10]. We call this problem *Heterogeneously Observable Imitation Learning* (HOIL).

There are two lines of research studying the related problems. The first line relates to domain adaptation: the observation space of the expert and the learner are the homogeneous, while some typical mismatches of distributions could exist: morphological mismatch, viewpoint mismatch, and dynamics mismatch [30, 17, 26]. However, these approaches are invalid when the observation spaces for experts and learners are completely different as in HOIL.

The second line studied IL under different observations similar to HOIL, and some representative works include Partially Observable Imitation Learning (POIL) [14, 36] and Learning by Cheating

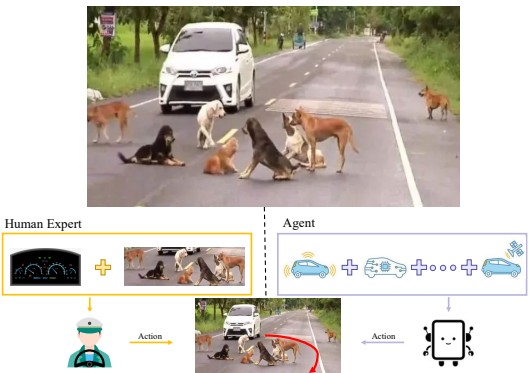

Figure 1: Autonomous driving: an example of the HOIL problem. Figures 1, 2 and 3 include some illustrations and pictures from the Internet (source: `www.vecteezy.com`).

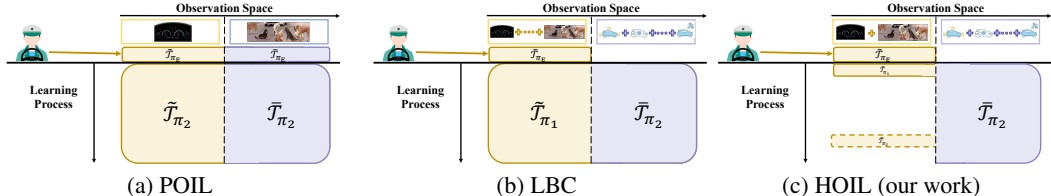

(a) POIL  (b) LBC  (c) HOIL (our work)

Figure 2: Comparisons of different IL processes under different observation spaces. The targets are all to learn $\pi_2$ based on the second observation space with an auxiliary policy $\pi_1$ from corresponding roll-out data $\widetilde{\mathcal{T}}$ and $\overline{\mathcal{T}}$. (a) POIL mainly emphasized that the expert can view full observations, while the observations for the learner are partial. (b) LBC assumed that the expert's observations contain more privileged information than the learner's. Both POIL and LBC can observe expert's observations all along. (c) HOIL limits the amount of expert's observations.

(LBC) [8], as depicted in Figure 2. Both POIL and LBC assume that the expert's observations can be easily accessed by the learner without any budget limit. However in practice, different from the learner observations, the access to expert's observations might be of high cost and invasive [6, 10], hindering the wide application of these methods.

In this paper, we initialize the study of the HOIL problem. We propose a learning process across observation spaces of experts and learners for solving this problem, and analyze the underlying issues of HOIL, i.e., the dynamics mismatch and the support mismatch. To tackle both two issues, we resort to the techniques of *importance-weighting* [12] and *learning with rejection* [9, 15] for active querying to propose the *Importance Weighting with REjection* (IWRE) approach. We evaluate the effectiveness of the IWRE algorithm in continuous control tasks of MuJoCo [33], and the challenging tasks of learning random access memory (RAM)-based policies given vision-based expert demonstrations in Atari [3] games. The results demonstrate that IWRE can significantly outperform existing IL algorithms in HOIL tasks, with limited access to expert observations.

## 2 Related Work

**Domain-Shifted IL.** For the standard IL process, where the learner and the expert share the same observation space, current state-of-the-art methods tend to learn the policy in an adversarial style [7], like GAIL [16]. When considering the domain mismatch problem, i.e., Domain-Shifted IL (DSIL), the research aims at addressing the *static distributional shift* of the optimal policies resulted from the environmental differences but still under homogeneous observation spaces. Stadie et al. [30], Sermanet et al. [29], and Liu et al. [23] studied the situation where the demonstrations are in view of a third person. Kim et al. [19] and Kim et al. [18] addressed the IL problem with morphological mismatch between the expert's and learner's environment. Stadie et al. [30], Tirinzoni et al. [32], and Desai et al. [11] focused on the calibration for the mismatch between simulators and the real world through some transfer learning styles. There are two major differences between HOIL and DSIL: One is that HOIL considers *heterogeneous* observation spaces instead of *homogeneous* ones; another

is that without observation heterogeneity, DSIL can directly align two fixed domains, which may not be realistic for solving HOIL when two observation spaces are totally different. Thus HOIL is a significantly more challenging problem than DSIL. Besides, Chen et al. [8] learned a vision-based agent from a privileged expert. But it can obtain expert's observations throughout the whole learning process, so it cannot handle the problem of the support mismatch under HOIL.

**POMDP.** The problem of POMDPs, in which only partial observations are available for the agent(s), has been studied in the context of multi-agent [25, 36] and imitation learning [14, 36] problems. But distinct from HOIL, in a POMDP, the learner only have partial observations and share a *same* underlying observation space with the expert, which would become an obstacle for them to make decisions correctly. For example, Warrington et al. [36] assumed that the observation of the learner is partial than that of the expert. Instead, in HOIL, expert's and learner's observations are totally *different* from each other, while the learner's observations are not belong to a part of the expert's. For HOIL, the main challenge is to deal with the mismatches between the observation spaces, especially when the access to expert's observations is strictly limited.

# 3 The HOIL Problem

In this section, we first give a formal definition of the HOIL setting, and then introduce the learning process for solving the HOIL problem.

## 3.1 Setting Definition

A HOIL problem is defined within a Markov decision process with mutiple observation spaces, i.e., $\langle \mathcal{S}, \{\mathcal{O}\}, \mathcal{A}, \mathcal{P}, \gamma \rangle$, where $\mathcal{S}$ denotes the state space, $\{\mathcal{O}\}$ denotes a set of observation spaces, $\mathcal{A}$ denotes the action space, $\mathcal{P} : \mathcal{S} \times \mathcal{A} \times \mathcal{S} \to \mathbb{R}$ denotes the transition probability distribution of the state and action, and $\gamma \in (0, 1]$ denotes the discount factor. Furthermore, a policy $\pi$ over an observation space $\mathcal{O}$ is defined as a function mapping from $\mathcal{O}$ to $\mathcal{A}$, and we denote by $\Pi_{\mathcal{O}}$ the set of all policies over $\mathcal{O}$. In HOIL, both the expert and the learner have their own observation spaces, which are denoted as $\mathcal{O}_{\mathrm{E}}$ and $\mathcal{O}_{\mathrm{L}}$ respectively. Both $\mathcal{O}_{\mathrm{E}}$ and $\mathcal{O}_{\mathrm{L}}$ are assumed to be produced by two bijective mappings $f_{\mathrm{E}} : \mathcal{S} \to \mathcal{O}_{\mathrm{E}}, f_{\mathrm{L}} : \mathcal{S} \to \mathcal{O}_{\mathrm{L}}$, which are unknown functions mapping the underlying true states to the observations. It is obvious to see that by this assumption, any policy over $\mathcal{O}_{\mathrm{E}}$ has a unique correspondence over $\mathcal{O}_{\mathrm{L}}$. This makes HOIL possible since the target of HOIL is to find the corresponding policy of the expert policy under $\mathcal{O}_{\mathrm{L}}$.

A state-action pair $(s, a)$, denoted by $x$, is called an *instance*. Also, a trajectory $\mathcal{T} = \{x_i\}, i \in [m]$ is a set of $m$ instances. For each observation space, $\widetilde{x} \in \widetilde{\mathcal{T}} \subseteq \mathcal{O}_{\mathrm{E}} \times \mathcal{A}$ and $\overline{x} \in \overline{\mathcal{T}} \subseteq \mathcal{O}_{\mathrm{L}} \times \mathcal{A}$, where $\mathcal{O}_{\mathrm{E}} = f_{\mathrm{E}}(\mathcal{S})$ and $\mathcal{O}_{\mathrm{L}} = f_{\mathrm{L}}(\mathcal{S})$. Furthermore, we define the *occupancy measure* of a policy $\pi$ under the state space $\mathcal{S}$ as $\rho_\pi : \mathcal{S} \times \mathcal{A} \to \mathbb{R}$ such that $\rho_\pi(x) = \pi(a|o)\Pr(o|s)\sum_{t=0}^{\infty} \gamma^t \Pr(s_t = s|\pi)$. Under HOIL, the learner accesses the expert demonstrations $\widetilde{\mathcal{T}}_{\pi_{\mathrm{E}}}$, a set of instances sampled from $\rho_{\pi_{\mathrm{E}}}$. The goal of HOIL is to learn a policy $\hat{\pi}$ as the corresponding policy of $\pi_{\mathrm{E}}$ over $\mathcal{O}_{\mathrm{L}}$. If $\mathcal{O}_{\mathrm{E}} = \mathcal{O}_{\mathrm{L}}$, HOIL degenerates to standard IL . GAIL [16] is one of the state-of-the-art IL approaches under this situation, which tries to minimize the divergence between the learner's and the expert's occupancy measures $d(\rho_{\hat{\pi}}, \rho_{\pi_{\mathrm{E}}})$. The objective of GAIL is

$$\min_{\hat{\pi}} \max_{w} \mathbb{E}_{x \sim \rho_{\pi_{\mathrm{E}}}} [\log D_w(\widetilde{x})] + \mathbb{E}_{x \sim \rho_{\hat{\pi}}} [\log(1 - D_w(\widetilde{x}))] - \mathbb{H}(\hat{\pi}), \tag{1}$$

where $\mathbb{H}(\hat{\pi})$ is the causal entropy performed as a regularization term, and $D_w : \mathcal{O}_{\mathrm{E}} \times \mathcal{A} \to [0, 1]$ is the discriminator of $\pi_{\mathrm{E}}$ and $\hat{\pi}$. GAIL solved Equation (1) by alternatively taking a gradient ascent step to train the discriminator $D_w$, and a minimization step to learn policy $\hat{\pi}$ based on an off-the-shelf RL algorithm with the pseudo reward $-\log D_w(\widetilde{x})$.

## 3.2 The Learning Process for Solving HOIL

In HOIL, we need to cope with the absence of the learner's observations in demonstrations and the high cost of collecting the expert's observations while learning. So we introduce a learning process with pretraining across two different observation spaces for solving HOIL, as abstracted in Figure 3.

**Pretraining.** Same to LBC [8], we assume that we can obtain an auxiliary policy $\pi_1$ based on $\mathcal{O}_{\mathrm{E}}$ at the beginning. $\pi_1$ can be directly provided by any sources, or trained by GAIL or behavior cloning as did in LBC. Besides, we use this $\pi_1$ to sample some data $\mathcal{T}_{\pi_1}$, which contain both observation under $\mathcal{O}_{\mathrm{E}}$ (i.e., $\widetilde{\mathcal{T}}_{\pi_1}$) and $\mathcal{O}_{\mathrm{L}}$ (i.e., $\overline{\mathcal{T}}_{\pi_1}$), in order to connect these two different observation spaces. We name $\mathcal{T}_{\pi_1} = \{\widetilde{\mathcal{T}}_{\pi_1}, \overline{\mathcal{T}}_{\pi_1}\}$ the *initial data*.

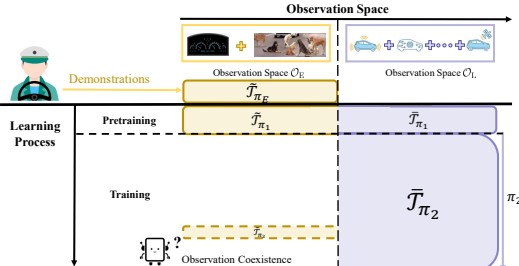

Figure 3: Illustration of a learning process across two different observation spaces for solving HOIL. $\pi_1$ is an auxiliary policy that additionally provided.

**Training.** Here we learn a policy $\pi_2$ from the initial data $\overline{\mathcal{T}}_{\pi_1}$ and the collected data $\overline{\mathcal{T}}_{\pi_2}$, under $\mathcal{O}_\mathrm{L}$ only. Besides, the learner is allowed for some operation of *observation coexistence* (OC): At some steps of learning, besides the observations $\mathcal{O}_\mathrm{L}$, the learner could also request $\widetilde{\mathcal{T}}_{\pi_2}$ from the corresponding observations $\mathcal{O}_\mathrm{E}$ (e.g., from the human-understandable sensors). The final objective of HOIL is to learn a good policy $\pi_2$ under $\mathcal{O}_\mathrm{L}$.

In practical applications, the auxiliary policy $\pi_1$ can also come from simulation training or direct imitation. But since $\pi_1$ is additionally provided, it is more practical to consider $\pi_1$ as a non-optimal policy. During training, OC is an essential operation for solving HOIL, which helps the learner address the issues of the dynamics mismatch and the support mismatch (especially the latter one). Also, in reality, we do not need an oracle for actions, which still needs OC for obtaining expert observations first, as in many active querying research [4, 8], so its cost will be relatively lower.

Besides, the related work [8] also required an initialized policy $\pi_1$ to solve their problem, which act as a teacher under privileged $\mathcal{O}_\mathrm{E}$ in the pretraining and then learned a vision-based student from the guidance of the teacher under both $\mathcal{O}_\mathrm{L}$ and $\mathcal{O}_\mathrm{E}$. Their setting can be viewed as a variety of HOIL with optimal $\pi_1$, unlimited $\mathcal{O}_\mathrm{E}$, and unlimited OC operations, so HOIL is actually a more practical learning framework.

## 4 Imitation Learning with Importance-Weighting and Rejection

In HOIL, the access frequency to $\mathcal{O}_\mathrm{E}$ is strictly limited, so it is unrealistic to learn $\pi_2$ in a Dataset Aggregation (DAgger) style [27] as in LBC. Therefore, we resort to learning $\pi_2$ with a learned reward function by inverse reinforcement learning [1] in an adversarial learning style [16, 13].

In addition, both $\mathcal{O}_\mathrm{E}$ and $\mathcal{O}_\mathrm{L}$ are assumed to share the same latent state space $\mathcal{S}$ as introduced in Section 3.1, so the following analysis will be based on $\mathcal{S}$, while the algorithm will handle the problem based on $\mathcal{O}_\mathrm{E}$ and $\mathcal{O}_\mathrm{L}$ specifically.

### 4.1 Dynamics Mismatch and Importance-Weighting

To analyze the learning process, we let $\rho_{\pi_\mathrm{E}}$, $\rho_{\pi_1}$, and $\rho_{\pi_2}$ be the occupancy measure distributions of the expert demonstrations, the initial data, and the data during training respectively. Since we need to consider the sub-optimality of $\pi_1$, $\rho_{\pi_1}$ should be a mixture distribution of the expert $\rho_{\pi_\mathrm{E}}$ and non-expert $\rho_{\pi_\mathrm{NE}}$, i.e., there exists some $\delta \in (0, 1)$ such that

$$\rho_{\pi_1} = \delta \rho_{\pi_\mathrm{E}} + (1 - \delta) \rho_{\pi_\mathrm{NE}}, \tag{2}$$

as depicted in Figure 4a. During training, the original objective of $\pi_2$ is to imitate $\pi_\mathrm{E}$ through demonstrations. To this end, the original objective of reward function $D_{w_2}$ for $\pi_2$ is to optimize

$$\max_{w_2} \mathbb{E}_{x \sim \rho_{\pi_2}} [\log D_{w_2}(\overline{x})] + \mathbb{E}_{x \sim \rho_{\pi_\mathrm{E}}} [\log(1 - D_{w_2}(\overline{x}))]. \tag{3}$$

But the expert demonstrations are only available under $\mathcal{O}_\mathrm{E}$. While during training, we can only utilize the initial data $\overline{\mathcal{T}}_{\pi_1} \sim \rho_{\pi_1}$ to learn $\pi_2$ and $D_{w_2}$. Besides, as $\pi_1$ is sub-optimal, directly imitating $\overline{\mathcal{T}}_{\pi_1}$ could reduce the performance of the optimal $\pi_2$ to that of $\pi_1$. So we use the importance-weighting to calibrate this dynamics mismatch, i.e.,

$$\max_{w_2} \mathcal{L}(D_{w_2}) = \mathbb{E}_{x \sim \rho_{\pi_2}} [\log D_{w_2}(\overline{x})] + \mathbb{E}_{x \sim \rho_{\pi_1}} [\alpha(x) \log(1 - D_{w_2}(\overline{x}))], \tag{4}$$

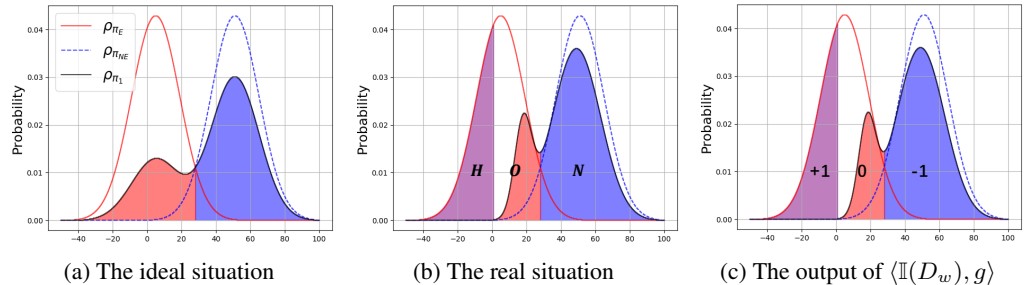

| (a) The ideal situation | (b) The real situation | (c) The output of $\langle \mathbb{I}(D_w), g \rangle$ |

Figure 4: The comparisons among the distributions of expert demonstrations $\rho_{\pi_\mathrm{E}}$, initial data $\rho_{\pi_1}$, and non-expert data $\rho_{\pi_\mathrm{NE}}$. The red and blue regions denote the expert and non-expert parts of $\rho_{\pi_1}$ respectively. $H$, $O$, and $N$ denote the latent demonstration, the observed demonstration, and the non-expert data respectively. (a) The ideal situation, where $\mathrm{supp}(\rho_{\pi_\mathrm{E}}) \setminus \mathrm{supp}(\rho_{\pi_1}) = \varnothing$; (b) The real situation, where $H := \mathrm{supp}(\rho_{\pi_\mathrm{E}}) \setminus \mathrm{supp}(\rho_{\pi_1}) \neq \varnothing$ in $\rho_{\pi_\mathrm{E}}$. (c) The target output of the combined model $\mathbb{I}[D_w^*]g^*$. The output $+1$, $0$, and $-1$ regions correspond to $H$, $O$, and $N$ respectively.

where $\alpha(x) \triangleq \frac{\rho_{\pi_\mathrm{E}}(x)}{\rho_{\pi_1}(x)}$ is an importance-weighting factor [12]. So the current issue lies in how to estimate $\frac{\rho_{\pi_\mathrm{E}}}{\rho_{\pi_1}}$ under $\mathcal{O}_\mathrm{E}$. To achieve this purpose, we need to bridge the expert demonstrations and the initial data. Therefore, here we use these two data sets to train an adversarial model $D_{w_1}$ in the same way as $D_{w_2}$ in the pretraining:

$$\max_{w_1} \mathcal{L}(D_{w_1}) \triangleq \mathbb{E}_{x \sim \rho_{\pi_1}}[\log D_{w_1}(\widetilde{x})] + \mathbb{E}_{x \sim \rho_{\pi_\mathrm{E}}}[\log(1 - D_{w_1}(\widetilde{x}))]. \tag{5}$$

If we write the training criterion (5) in the form of integral, i.e.,

$$\max_{w_1} \mathcal{L}(D_{w_1}) = \int_x [\rho_{\pi_1} \log D_{w_1} + \rho_{\pi_\mathrm{E}} \log(1 - D_{w_1})]dx, \tag{6}$$

then, by setting the derivative of the objective (6) to 0 ($\frac{\partial \mathcal{L}}{\partial D_{w_1}} = 0$), we can obtain the optimum $D_{w_1}$:

$$D_{w_1}^* = \frac{\rho_{\pi_1}}{\rho_{\pi_1} + \rho_{\pi_\mathrm{E}}}, \tag{7}$$

in which the order of differentiation and integration was changed by the Leibniz rule. Besides, we can sufficiently train $D_{w_1}$ using the initial data $\widetilde{\mathcal{T}}_{\pi_1}$ and the expert demonstrations $\widetilde{\mathcal{T}}_{\pi_\mathrm{E}}$. Then $D_{w_1}$ will be good enough to estimate the importance-weighting factor, i.e.,

$$\alpha(x) \triangleq \frac{\rho_{\pi_\mathrm{E}}}{\rho_{\pi_1}} = \frac{1 - D_{w_1}^*(\widetilde{x})}{D_{w_1}^*(\widetilde{x})} \approx \frac{1 - D_{w_1}(\widetilde{x})}{D_{w_1}(\widetilde{x})}. \tag{8}$$

In this way, we can use $D_{w_1}$, which can connect demonstrations and initial data, to calibrate the learning process of $D_{w_2}$. The final optimization objective for $D_{w_2}$ is

$$\max_{w_2} \mathcal{L}(D_{w_2}) = \mathbb{E}_{x \sim \rho_{\pi_2}} \log D_{w_2}(\overline{x}) + \mathbb{E}_{x \sim \rho_{\pi_1}} \frac{1 - D_{w_1}(\widetilde{x})}{D_{w_1}(\widetilde{x})} \log[1 - D_{w_2}(\overline{x})]. \tag{9}$$

In this way, $D_{w_2}$ can effectively dig out the expert part of $\rho_{\pi_1}$ and produce efficient rewards for $\pi_2$.

## 4.2 Support Mismatch

So far the challenges have still been similar to homogeneously observable imitation learning. However, our preliminary experiments demonstrated that merely importance-weighting is not enough to fix the problem that occurred by the absence of interactions under $\mathcal{O}_\mathrm{E}$. So there exist some other issues between the expert demonstrations and the initial data. To find out the underlying issues, we plot the t-Distributed Stochastic Neighbor Embedding (t-SNE) [34] visualizations of these two empirical distributions under $\mathcal{O}_\mathrm{E}$ on *Hopper* and *Walker2d*, as shown in Figure 5. Twenty trajectories were collected for both the expert demonstrations and the initial data. We can observe that there exist some

high-density regions of demonstrations in which the initial data do not cover; that is, there exist some regions of the demonstrations that $\pi_1$ did *not explore*. Wang et al. [35] found a similar phenomenon in the standard IL setting. On the other hand, the importance-weighting $\alpha$ cannot calibrate this situation where $\frac{\rho_{\pi_E}}{\rho_{\pi_1}} = \infty$.

To formulate this problem, here we introduce the *Support Set* of the occupancy measure:

**Definition 1** (Support Set). *The support set of a occupancy measure $\rho$ is the subset of the domain containing the elements which are not mapped to zero:*

$$\text{supp}(\rho) \coloneqq \{x \in \mathcal{S} \times \mathcal{A} | \rho(x) \neq 0\}. \tag{10}$$

Due to the sub-optimality of $\pi_1$, $\text{supp}(\rho_{\pi_E}) \backslash \text{supp}(\rho_{\pi_1}) \neq \varnothing$ (see Figure 4b). We call this part the *Latent Demonstration*, defined as:

**Definition 2** (Latent Demonstration). *The latent demonstration $H$ is the set of those $x \in \mathcal{S} \times \mathcal{A}$ that belong to the relative complement of $\text{supp}(\rho_{\pi_1})$ in $\text{supp}(\rho_{\pi_E})$:*

(a) Hopper  (b) Walker2d

Figure 5: t-SNE visualizations of expert demonstrations and collected data of $\pi_1$ under $\mathcal{O}_E$.

$$H \coloneqq \{x \in \mathcal{S} \times \mathcal{A} | \text{supp}(\rho_{\pi_E}) \setminus \text{supp}(\rho_{\pi_1})\}. \tag{11}$$

Also, another part of the demonstration is named the *Observed Demonstration*, defined as:

**Definition 3** (Observed Demonstration). *The observed demonstration $O$ is the set of those $x \in \mathcal{S} \times \mathcal{A}$ that belong to the complement of $H$ in $\text{supp}(\rho_{\pi_E})$:*

$$O \coloneqq \{x \in \mathcal{S} \times \mathcal{A} | \text{supp}(\rho_{\pi_E}) \cap \text{supp}(\rho_{\pi_1})\}. \tag{12}$$

Besides, the data outside of demonstrations should be non-expert data:

**Definition 4** (Non-Expert Data). *The non-expert data $N$ is the set of those $x \in \mathcal{S} \times \mathcal{A}$ that out of $\text{supp}(\rho_{\pi_E})$:*

$$N \coloneqq \{x \in \mathcal{S} \times \mathcal{A} | \rho_{\pi_E}(x) = 0\}. \tag{13}$$

In other words, the sub-optimality of $\pi_1$ will cause not only the dynamics mismatch, but also the appearance of the latent demonstration $H$. We call the latter one the problem of *Support Mismatch*. Intuitively, *when $\pi_2 \to \pi_E$, we have $H \to \varnothing$, monotonously*. So in order to fix the support mismatch between $\rho_{\pi_E}$ and $\rho_{\pi_1}$, *guiding $\pi_2$ to find out $H$ is the key*.

In addition, the support mismatch problem can be viewed as an inverse problem of the Out Of Distribution (OOD) problem that frequently occurred in offline RL setting [21], in which they tried to avoid $\text{supp}(\rho_{\pi_1}) \setminus \text{supp}(\rho_{\pi_E})$ instead.

### 4.3 Imitation Learning with Rejection

We can observe that $H \cup O \cup N = \mathcal{S} \times \mathcal{A}$. So it is desirable to filter out $H$ from $O$ and $N$. Meanwhile, $D_{w_1}$ and $D_{w_2}$ can only classify $O \cup H$ and $N$, under $\mathcal{O}_E$ and $\mathcal{O}_L$ respectively. Therefore, here we design two models $g_1 : \mathcal{O}_E \times \mathcal{A} \to \{0, 1\}$ and $g_2 : \mathcal{O}_L \times \mathcal{A} \to \{0, 1\}$ (Output 0: $x \in O$ and output 1: otherwise), so that given $x \sim \mathcal{T}$ (corresponding $\widetilde{x} \sim \widetilde{\mathcal{T}}$ and $\overline{x} \sim \overline{\mathcal{T}}$) they can satisfy:

$$H = \{x \in \mathcal{S} \times \mathcal{A} | \mathbb{I}[D_{w_1}^*(\widetilde{x})]g_1^*(\widetilde{x}) = \mathbb{I}[D_{w_2}^*(\overline{x})]g_2^*(\overline{x}) = +1\}, \tag{14}$$

$$O = \{x \in \mathcal{S} \times \mathcal{A} | \mathbb{I}[D_{w_1}^*(\widetilde{x})]g_1^*(\widetilde{x}) = \mathbb{I}[D_{w_2}^*(\overline{x})]g_2^*(\overline{x}) = 0\}, \tag{15}$$

$$N = \{x \in \mathcal{S} \times \mathcal{A} | \mathbb{I}[D_{w_1}^*(\widetilde{x})]g_1^*(\widetilde{x}) = \mathbb{I}[D_{w_2}^*(\overline{x})]g_2^*(\overline{x}) = -1\}, \tag{16}$$

respectively, where $\mathbb{I}[\cdot]$ takes $+1$ if $\cdot > 0.5$, and $-1$ otherwise. The target combined model $\mathbb{I}[D_w^*(x)]g^*(x)$ is depicted in Figure4c.

To this end, both $g_1$ and $g_2$ should be able to cover $O$, meanwhile $g_2$ can be adaptive to continuously change of $\rho_{\pi_2}$ due to the update of $\pi_2$. Here we learn $g_1$ and $g_2$ in a rejection form, to *reject $O$ from*

$O \cup H$ (where $\mathbb{I}(D_w) = +1$). Concretely, the rejection setting is the same as that in Cortes et al. [9]. Also inspired by Geifman et al. [15], the optimization objective of the combination of $D_w$ and $g$ is

$$\mathcal{L}(D_w, g) \triangleq \hat{l}(D_w, g) + \lambda \max(0, c - \hat{\phi}(g))^2, \tag{17}$$

where $c > 0$ denotes the target coverage, and $\lambda$ denotes the factor for controlling the relative importance of rejection. Besides, the empirical coverage $\hat{\phi}(g)$ is defined as

$$\hat{\phi}(g|X) \triangleq \frac{1}{m} \sum_{i=1}^{m} g(x_i), \tag{18}$$

where a batch of data $X = \{x_i\}, i \in [m]$. The empirical rejection risk $\hat{l}(D_w, g)$ is the ratio between the covered risk of the discriminator and the empirical coverage:

$$\hat{l}(D_w, g) \triangleq \frac{\frac{1}{m} \sum_{i=1}^{m} \langle \mathcal{L}(D_w(x_i)), g(x_i) \rangle}{\hat{\phi}(g)}. \tag{19}$$

Meanwhile, both $D_{w_1}$ and $g_1$ can access $\rho_{\pi_{\mathrm{E}}}$ under $\mathcal{O}_{\mathrm{E}}$ directly. So given $\overline{x} \sim \overline{\mathcal{T}}_{\pi_2}$ under $\mathcal{O}_{\mathrm{L}}$, once $\langle \mathbb{I}(D_{w_2}(\overline{x})), g_2(\overline{x}) \rangle = +1$, we can query the corresponding observations $\widetilde{x}$ of $\overline{x}$ through OC operation and use $\langle \mathbb{I}(D_{w_1}(\widetilde{x})), g_1(\widetilde{x}) \rangle$ to calibrate the output of $g_2$ and $D_{w_2}$. In this way, $g_2$ and $D_{w_2}$ can be entangled together and adaptively guide $\pi_2$ to find out the latent demonstrations $H$ under $\mathcal{O}_{\mathrm{L}}$.

### 4.4 IWRE

Here we combine the importance-weighting and rejection into a unified whole, to propose a novel algorithm named Importance Weighting with REjection (IWRE). Concretely, in a HOIL process:

**Pretraining.** We train a discriminator $D_{w_1}$ by Equation (5) and its corresponding rejection model $g_1$ by Equation (17) using the initial data and the expert demonstrations.

**Training.** We train a discriminator $D_{w_2}$ by the combination of Equation (9) and Equation (17), as well as its corresponding rejection model $g_2$ by Equation (17), using the initial data, the data collected by $\pi_2$, and the output of $D_{w_1}$ with $g_1$ through OC operation. Also, $\pi_2$ will be updated with $D_{w_2}$ and $g_2$ asymmetrically as in GAIL.

The pseudo-code of our algorithm is provided in the supplementary material.

## 5 Experiment

In this section, we validate our algorithm in Atari 2600 [3] (GPL License) and MuJoCo [33] (Academic License) environments. The experiments were designed to investigate:

1) Can IWRE achieve significant performance under HOIL tasks?

2) Can IWRE deal with the support mismatch problem?

3) During training, is active querying for HOIL indeed necessary?

Below we first introduce the experimental setup and then investigate the above questions. More results and experimental details are included in the supplementary material.

### 5.1 Experimental Setup

**Environments.** We choose three pixel-memory based games in Atari and five continuous control objects in MuJoCo on OpenAI platform [5] (MIT License). Details as below:

1. **Pixel-memory Atari games.** $\mathcal{O}_{\mathrm{E}}$: $84 \times 84 \times 4$ raw pixels; $\mathcal{O}_{\mathrm{L}}$: 128-byte random access memories (RAM). Expert: converged DQN-based agents [24]. Atari games contain two totally isolated views: raw pixels and RAM, under the same state. Through these environments, we want to investigate whether the agent can learn an effective policy from demonstrations under completely different observation spaces. Moreover, IL with visual observations only is already very difficult [7], while learning a RAM-based policy can be even more challenging [3, 31], so few IL research reported desirable results on this task.

2. **Continuous control MuJoCo objects.** $\mathcal{O}_{\mathrm{E}}$: half of original observation features; $\mathcal{O}_{\mathrm{L}}$: another half of original observation features. Expert: converged DDPG-based agents [22]. The

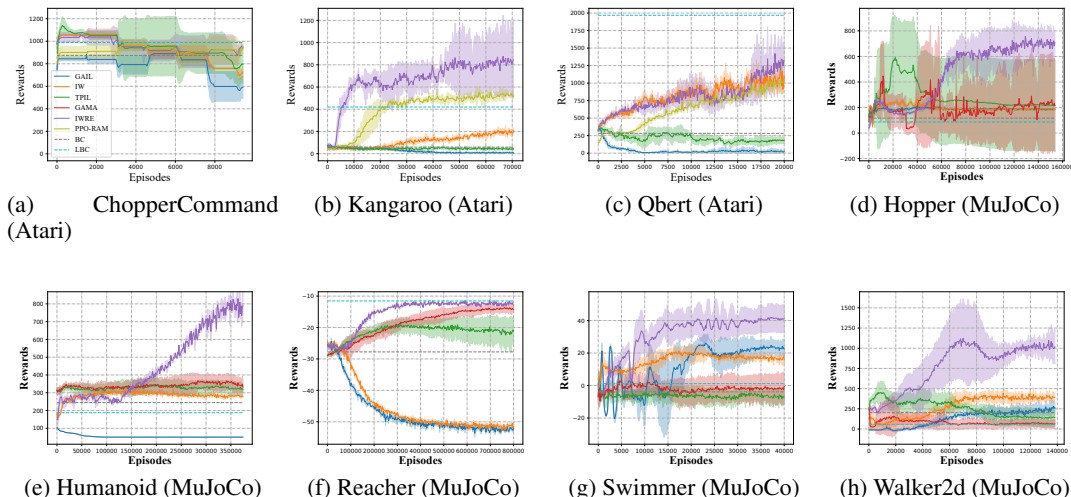

(a)      ChopperCommand   (b) Kangaroo (Atari)   (c) Qbert (Atari)   (d) Hopper (MuJoCo)
(Atari)

(e) Humanoid (MuJoCo)   (f) Reacher (MuJoCo)   (g) Swimmer (MuJoCo)   (h) Walker2d (MuJoCo)

Figure 6: The learning curves of each method, where the shaded region indicates the standard deviation.

features of MuJoCo contain monotonous information like the direction, position, velocity, etc., of an object. Through these environments, we want to investigate whether the agent can learn from demonstrations with complementary signals under observations with missing information. Meanwhile, we make sure RL algorithms can obtain comparable performances under $\mathcal{O}_E$ and $\mathcal{O}_L$. More details are reported in the supplementary material.

Besides, twenty expert trajectories were collected for each environment. Each result contains five trials with different random seeds. All experiments were conducted on server clusters with NVIDIA Tesla V100 GPUs. The summary of the environments is gathered in the supplementary material.

**Baselines.** Six basic contenders were included in the experiments: Vanilla **GAIL** [16], GAIL with importance-weighting [12] (**IW**), third-person IL [30] (**TPIL**), generative adversarial MDP alignment [19] (**GAMA**), behavioral cloning [2] (**BC**), and learning by cheating [8] (**LBC**). For IW, we utilized the discriminator $D_{w_1}$ trained in the pretraining to calculate the importance weight; also the optimization objective for $D_{w_2}$ during training is the same as Equation (9); TPIL learns the third-person demonstrations by leading the cross-entropy loss into the update of the feature extractor; GAMA learns a mapping function $\psi$ in view of adversarial training to align the observation of the target domain into the source domain, and thereby can utilize the policy in the source domain for zero-shot imitation. For fairness, we allowed the interaction between the policy and the environment for GAMA under HOIL; LBC uses $\pi_1$ learned from privileged states as a teacher to train $\pi_2$ in a DAgger [27] style, so here we allowed LBC to access $\mathcal{O}_E$ during the whole IL process. In Atari, to investigate whether our method could achieve good performance for RAM-based control, we further included a contender **PPO-RAM**, which uses proximal policy optimization (PPO) [28] to perform RL directly with environmental true rewards under the RAM-based observations. More detailed setup including query strategies for TPIL and GAMA, network architecture, and hyper-parameters are reported in the supplementary material.

**Learning process.** To simulate the situation that $\mathcal{O}_E$ is costly, the steps for training $\pi_1$ was set as 1/4 of that for training $\pi_2$, using GAIL [16]/HashReward [7] under the $\mathcal{O}_E$ space for MuJoCo/Atari environments. The learning steps were $10^7$ for MuJoCo and $5 \times 10^6$ for Atari environments. In the pretraining, we sampled 20 trajectories from $\pi_1$, and the data from each trajectory had both $\mathcal{O}_E$ and $\mathcal{O}_L$ observations. In the training, each method learned $4 \times 10^7$ steps for MuJoCo and $2 \times 10^7$ steps for Atari under the $\mathcal{O}_L$ space to obtain $\pi_2$.

## 5.2  Results

Experimental results are reported in Figure 6. Since the mapping function is hard to learn when input is RAM and output is raw images, we omit the results of GAMA in Atari. We can observe that while IW is better than GAIL in most environments, both GAIL and IW can hardly outperform $\pi_1$.

Because they just imitated the performance of $\pi_1$ instead of $\pi_E$, even with importance-weighting for calibration. For TPIL, its learning process was extremely unstable on *Hopper*, *Swimmer*, and *Walker2d* due to the continuous distribution shift. Furthermore, the performance of GAMA was not satisfactory in *Hopper* and *Walker2d* because its mapping function is hard to learn well when the support mismatch appears. The results of TPIL and GAMA demonstrate that DSIL methods will be invalid under heterogeneous observations as in HOIL tasks. On Atari environments, $\mathcal{O}_E$ contains more privileged information than $\mathcal{O}_L$, so LBC can achieve good performance. But when $\mathcal{O}_E$ is not more privileged than $\mathcal{O}_L$, like in most environments of MuJoCo, its performance will decrease due to the support mismatch, which would make it even worse than BC. Finally, IWRE obtained the best performance on 6/8 environments, and comparable performance with LBC on *Reacher*, which shows the effectiveness of our method even with limited access to $\mathcal{O}_E$ (LBC can access to $\mathcal{O}_E$ all the time). Besides, we can see that the performance differences between the GAIL/IW and IWRE/TPIL/GAMA/LBC are huge (especially on *Reacher*) because of the absence of queries, which demonstrates that the query operation is indeed necessary for HOIL problems.

Moreover, even learned with true rewards, PPO-RAM surprisingly failed to achieve comparable performance to IWRE, which shows that IWRE could possibly learn more effective rewards than true environmental rewards in RAM-input tasks. The results verify that, IWRE provides a powerful approach for tackling HOIL problems, even under the situation that the demonstrations are gathered from such a different observation space, meanwhile $\mathcal{O}_E$ is strictly limited during training.

**t-SNE visualization of $\rho_{\pi_2}$ and $\rho_{\pi_E}$ under $\mathcal{O}_E$.** In Section 4.2, we point that the sub-optimality of $\pi_1$ will cause the problem of support mismatch, which is embodied as the appearance of the latent demonstration $H$ during training. Also the empirical results in Figure 5 on *Hopper* and *Walker2d* verify the existence of this problem. So we want to investigate whether the superiority of IWRE indeed comes from successfully tackling the support mismatch problem. To this end, we plotted the t-SNE visualization of the same expert demonstrations as in Section 4.2 and the collected data of $\pi_2$ by IWRE under $\mathcal{O}_E$ ($\mathcal{O}_E$ is hidden to $\pi_2$). All setups are the same as in Section 4.2. From the results shown in Figure 7, we can see that even under $\mathcal{O}_E$, which cannot be obtained by $\pi_2$, almost all high-density regions of the demonstrations were covered by the collected data. Meanwhile, the latent demonstration $H$ is dug out nearly. The results demonstrate that IWRE basically solves the problem of support mismatch and thereby performs well in these environments.

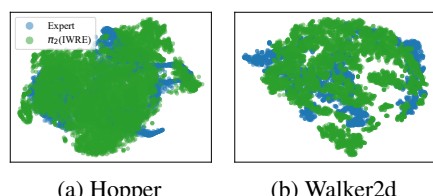

(a) Hopper  (b) Walker2d

Figure 7: t-SNE visualizations of expert demonstrations and collected data of $\pi_2$ under $\mathcal{O}_E$. The high-density regions of the expert demonstrations were covered by the collected data of $\pi_2$ of IWRE.

Besides, some collected data of $\pi_2$ of IWRE were out of the distribution of the demonstrations, which means $\pi_2$ slightly overly explored the environment. Since $\mathcal{O}_E$ is hidden to $\pi_2$, the reward function will encourage $\pi_2$ to explore more areas to fix the support mismatch problem. Meanwhile, the out-of-distribution problem in HOIL is not as severe as in the offline RL settings [21], so this over-exploration phenomenon makes sense.

## 6 Conclusion

In this paper, we proposed a new learning framework named *Heterogeneously Observable Imitation Learning* (HOIL), to formulate the situations where the observation space of demonstrations is different from that of the imitator while learning. We formally modeled a learning process of HOIL, in which the access to the observations of an expert is limited due to the high cost. Furthermore, we analyzed underlying challenges during training, i.e., the dynamics mismatch and the support mismatch, on the occupancy distributions between the demonstrations and the policy. To tackle these challenges, we proposed a new algorithm named Importance Weighting with REjection (IWRE), using importance-weighting and learning with rejection. Experimental results showed that the direct imitation and domain adaptive methods could not solve this problem, while our approach obtained promising results. In the future, we hope to involve the theoretical guarantee for our algorithm IWRE and investigate how many $\mathcal{O}_E$ do we need to query to learn a promising $\pi_2$. Furthermore, we hope to use the learning framework of HOIL and IWRE to tackle more learning scenarios with demonstrations in different spaces.

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
