# Supplementary Materials of
# Seeing Differently, Acting Similarly:
# Heterogeneously Observable Imitation Learning

## 1 Algorithm

The pseudo codes of our algorithm are illustrated in Algorithms 1 and 2.

## 2 Definitions

The core challenges of HOIL, i.e., dynamics mismatch and support mismatch, are illustrated as below.

**Definition 1** (Dynamics Mismatch). *The dynamics mismatch between the demonstrations and the initial data denotes the situation that:*

$$\frac{\rho_{\pi_\mathrm{E}}}{\rho_{\pi_1}} = \frac{\pi_\mathrm{E}(a|o) \sum_{t=0}^{\infty} \gamma^t \mathrm{Pr}(s_t = s|\pi_\mathrm{E})}{\pi_1(a|o) \sum_{t=0}^{\infty} \gamma^t \mathrm{Pr}(s_t = s|\pi_1)} \neq 1. \tag{1}$$

**Definition 2** (Support Mismatch). *The support mismatch between the demonstrations and the initial data denotes the situation that:*

$$\mathrm{supp}(\rho_{\pi_\mathrm{E}}) \setminus \mathrm{supp}(\rho_{\pi_1}) = \{x \in \mathcal{S} \times \mathcal{A} | \rho_{\pi_\mathrm{E}}(x) \neq 0\} \setminus \{x \in \mathcal{S} \times \mathcal{A} | \rho_{\pi_1}(x) \neq 0\} \neq \varnothing. \tag{2}$$

Table 1: Environmental summary of the tasks.

| Environment | Observation Space $\mathcal{O}_\mathrm{E}$ | Observation Space $\mathcal{O}_\mathrm{L}$ | Expert Rewards |
|---|---|---|---|
| Qbert | | | $4750.00 \pm 50.51$ |
| ChopperCommand | $84 \times 84 \times 4$(image) | 128(unsigned int) | $3135.00 \pm 145.86$ |
| Kangaroo | | | $4175.00 \pm 94.21$ |
| Hopper | 8 | 9 | $709.96 \pm 75.54$ |
| Humanoid | 4 | 4 | $539.20 \pm 26.26$ |
| Reacher | 5 | 6 | $-8.99 \pm 0.54$ |
| Swimmer | 5 | 6 | $52.24 \pm 1.29$ |
| Walker2d | 188 | 188 | $929.97 \pm 24.09$ |

## 3 Detailed Setup for the Experiments

The details of the environments are reported in Table 1. Also, the detailed comparisons of the contenders (both in the main paper and the supplementary material) and IWRE are gathered in Table 2.

Submitted to 36th Conference on Neural Information Processing Systems (NeurIPS 2022). Do not distribute.

---

**Algorithm 1** `IWRE.Pretraining`

---

**Input:** Auxiliary policy $\pi_1$; Expert demonstrations $\widetilde{\mathcal{T}}_{\pi_E}$.
**Output:** Evolving data $\{\widetilde{\mathcal{T}}_{\pi_1}, \overline{\mathcal{T}}_{\pi_1}\}$; Discriminator $D_{w_1}$; Rejection model $g_1$.
 1: **function** IWRE.PRETRAINING($\pi_1$)
 2:     Sample the evolving data $\{\widetilde{\mathcal{T}}_{\pi_1}, \overline{\mathcal{T}}_{\pi_1}\} \sim \rho_{\pi_1}$ by $\pi_1$.
 3:     Train $D_{w_1}$ and $g_1$ by Equation (5) and (17) respectively using $\widetilde{\mathcal{T}}_{\pi_E}$ and $\widetilde{\mathcal{T}}_{\pi_1}$.
 4:     **return** $\overline{\mathcal{T}}_{\pi_1}, D_{w_1}, g_1$
 5: **end function**

---

---

**Algorithm 2** `IWRE.Training`

---

**Input:** Expert demonstrations $\widetilde{\mathcal{T}}_{\pi_E}$; Evolving data $\overline{\mathcal{T}}_{\pi_1}$; Discriminator $D_{w_1}$; Rejection model $g_1$.
**Output:** Target policy $\pi_2$.
 1: **function** IWRE.TRAINING($\widetilde{\mathcal{T}}_{\pi_E}, \overline{\mathcal{T}}_{\pi_1}, D_{w_1}, g_1$)
 2:     Initialize $\pi_2$, $D_{w_2}$, and $g_2$.
 3:     **for** each step $t$ **do**
 4:         Sample $\overline{\mathcal{T}}_{\pi_2} \sim \rho_{\pi_2}$ by $\pi_2$.
 5:         **for** each mini-batch $\{\overline{x}_{\pi_2}\}$ and $\{\overline{x}_{\pi_1}\}$ from $\overline{\mathcal{T}}_{\pi_2}$ and $\overline{\mathcal{T}}_{\pi_1}$ **do**
 6:             Update $\pi_2$ by RL algorithms (such as PPO [4]) using instances $\{\overline{x}_{\pi_2}\}$ and pseudo rewards $\{-\log D_{w_2}(\overline{x}_{\pi_2})\}$.
 7:             Update $D_{w_2}$ by Equation (9) using negative instances $\{\overline{x}_{\pi_2}\}$ and positive ones $\{\overline{x}_{\pi_1}\}$.
 8:             **if** $\langle \mathbb{I}(D_{w_2}(\overline{x}_{\pi_2})), g_2(\overline{x}_{\pi_2}) \rangle = +1$ **then**
 9:                 Query the $\mathcal{O}_E$ observation of $\overline{x}_{\pi_2}$, i.e., $\widetilde{x}_{\pi_2}$, through OC operation.
10:                 Update $D_{w_2}$ and $g_2$ by Equation (17) using the instance $\overline{x}_{\pi_2}$ and the corresponding label $\langle \mathbb{I}(D_{w_1}(\widetilde{x}_{\pi_2})), g_1(\widetilde{x}_{\pi_2}) \rangle$.
11:             **end if**
12:         **end for**
13:     **end for**
14:     **return** $\pi_2$
15: **end function**

---

About query strategies, for TPIL and GAMA, if the output of the domain invariant discriminator is larger than 0.5, which means the encoder fails to generate proper features to confuse its discriminator, then we would query $\mathcal{O}_E$ of this data to update the encoder. For IWRE, the threshold of the rejection model $g$ and the discriminator $D_{w_2}$ was also 0.5, which means that if $g_2(\overline{x}) > 0.5$ meanwhile $D_{w_2}(\overline{x}) > 0.5$, $\mathcal{O}_E$ of this data would be queried. $D_{w_2}$, $\pi_2$, and the encoder (for TPIL/GAMA) were pretrained for 100 epochs for all methods using evolving data during pretraining. The basic RL algorithm is PPO, and the reward signals of all methods were normalized into $[0, 1]$ to enhance the performance of RL [2]. The buffer size for TPIL and IWRE was set as 5000. Each time the buffer is full, the encoder and the rejection model will be updated for 4 epochs; also LBC will update $\pi_2$ for 100 epochs with batch size 256 using the cross-entropy loss for Atari and the mean-square loss for MuJoCo. We set all hyper-parameters, update frequency, and network architectures of the policy part the same as Dhariwal et al. [2]. Besides, the hyper-parameters of the discriminator for all methods were the same: The rejection model and discriminator were updated using Adam with a decayed learning rate of $3 \times 10^{-4}$; the batch size was 256. The ratio of update frequency between the learner and discriminator was 3: 1. The target coverage $c$ in Equation (17) was set as 0.8. $\lambda$ in Equation (17) was 1.0.

## 4   RL Performance under the Divisions of MuJoCo

Here we report the performance under the division of $\mathcal{O}_E$ and $\mathcal{O}_L$ in MuJoCo. The details of the division are reported in Table 3. We use DDPG-based [3] agent with $10^7$ training steps and repeat 10 times with different random seeds. The results are shown in Figure 1. We can see that the agent can obtain comparable performance under $\mathcal{O}_E$ and $\mathcal{O}_L$. So for MuJoCo environments, the fairness of the division in HOIL can be guaranteed, and $\mathcal{O}_E$ is not more or less privileged than $\mathcal{O}_L$.

Table 2: Comparisons between all contenders and IWRE in HOIL.

| Algorithm | Considering heterogeneous observations | Being able to query | Not requiring $\mathcal{O}_E$ all along |
|---|---|---|---|
| GAIL | ✗ | ✗ | ✓ |
| GAIL-Rand | ✗ | ✓ | ✓ |
| IW | ✗ | ✗ | ✓ |
| IW-Rand | ✗ | ✓ | ✓ |
| TPIL | ✗ | ✓ | ✓ |
| GAMA | ✗ | ✓ | ✓ |
| BC | ✗ | ✗ | ✓ |
| LBC | ✓ | ✓ | ✗ |
| PPO-RAM | ✗ | ✗ | ✓ |
| IWRE | ✓ | ✓ | ✓ |

Table 3: The observation division into $\mathcal{O}_E$ and $\mathcal{O}_L$ in MuJoCo. The numbers denote the randomly selected observation indexes in the corresponding MuJoCo environment on OpenAI Gym [1] platform.

| | $\mathcal{O}_E$ | $\mathcal{O}_L$ |
|---|---|---|
| Walker2d | [5, 7, 8, 10, 11, 14, 15, 16] | [0, 1, 2, 3, 4, 6, 9, 12, 13] |
| Swimmer | [0, 3, 6, 7] | [1, 2, 4, 5] |
| Reacher | [0, 1, 7, 8, 10] | [2, 3, 4, 5, 6, 9] |
| Hopper | [1, 3, 6, 7, 9, 10] | [0, 2, 4, 5, 8] |
| Humanoid | [2, 3, 5, 6, 7, 10, 11, 12, 13, 16, 18, 19, 22, 23, 25, 29, 31, 32, 34, 36, 37, 40, 43, 44, 45, 47, 48, 49, 51, 54, 56, 57, 61, 63, 65, 66, 67, 68, 77, 78, 82, 86, 87, 89, 90, 93, 94, 95, 97, 98, 99, 102, 103, 108, 110, 112, 113, 117, 119, 120, 121, 122, 123, 124, 126, 127, 128, 133, 135, 144, 146, 147, 148, 151, 152, 153, 158, 160, 161, 162, 166, 167, 170, 171, 173, 174, 176, 177, 178, 180, 181, 184, 185, 187, 188, 191, 194, 198, 199, 200, 201, 202, 207, 208, 209, 210, 211, 212, 214, 215, 219, 223, 227, 228, 229, 231, 232, 233, 234, 236, 237, 238, 242, 244, 246, 248, 251, 253, 257, 258, 259, 260, 262, 264, 265, 267, 268, 271, 272, 273, 275, 278, 279, 280, 281, 285, 287, 289, 290, 291, 293, 294, 296, 299, 302, 304, 305, 306, 307, 308, 311, 312, 313, 315, 316, 319, 322, 326, 328, 329, 332, 337, 342, 343, 344, 345, 349, 358, 361, 362, 364, 365, 366, 368, 370, 372, 373, 375] | [0, 1, 4, 8, 9, 14, 15, 17, 20, 21, 24, 26, 27, 28, 30, 33, 35, 38, 39, 41, 42, 46, 50, 52, 53, 55, 58, 59, 60, 62, 64, 69, 70, 71, 72, 73, 74, 75, 76, 79, 80, 81, 83, 84, 85, 88, 91, 92, 96, 100, 101, 104, 105, 106, 107, 109, 111, 114, 115, 116, 118, 125, 129, 130, 131, 132, 134, 136, 137, 138, 139, 140, 141, 142, 143, 145, 149, 150, 154, 155, 156, 157, 159, 163, 164, 165, 168, 169, 172, 175, 179, 182, 183, 186, 189, 190, 192, 193, 195, 196, 197, 203, 204, 205, 206, 213, 216, 217, 218, 220, 221, 222, 224, 225, 226, 230, 235, 239, 240, 241, 243, 245, 247, 249, 250, 252, 254, 255, 256, 261, 263, 266, 269, 270, 274, 276, 277, 282, 283, 284, 286, 288, 292, 295, 297, 298, 300, 301, 303, 309, 310, 314, 317, 318, 320, 321, 323, 324, 325, 327, 330, 331, 333, 334, 335, 336, 338, 339, 340, 341, 346, 347, 348, 350, 351, 352, 353, 354, 355, 356, 357, 359, 360, 363, 367, 369, 371, 374] |

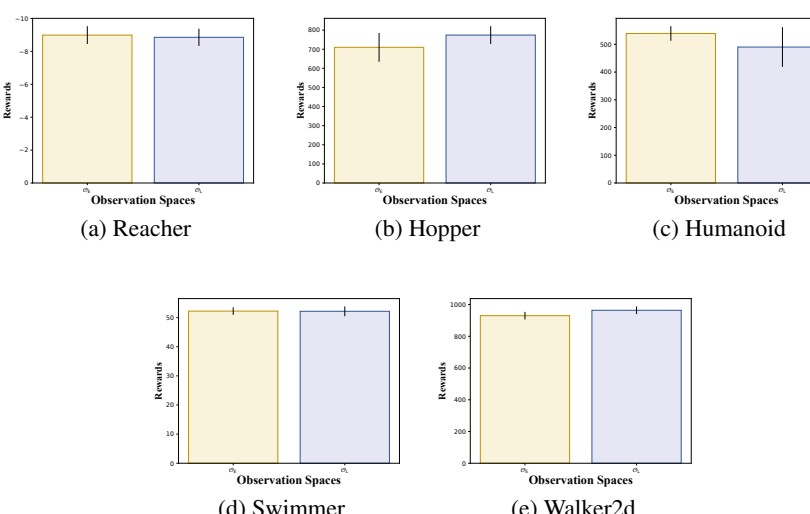

(a) Reacher     (b) Hopper     (c) Humanoid

(d) Swimmer     (e) Walker2d

Figure 1: The performance of RL methods under the division of $\mathcal{O}_E$ and $\mathcal{O}_L$ in MuJoCo. The agent can obtain comparable performances under $\mathcal{O}_E$ and $\mathcal{O}_L$, so that we can make sure the fairness of the experiment of HOIL in the main paper.

38

## 5 Estimation of $H$, $O$, and $N$ by $\mathbb{I}[D_{w_2}]g_2$

To investigate the ability of IWRE to distinguish the areas of latent demonstrations $H$, observed demonstrations $O$, and non-expert data $N$ during policy learning, we recorded the accuracy and estimated percentage of each area on *Hopper* and *Walker2d*. The calculations of each curve are shown as below:

$$\text{Accuracy\_}H = \frac{\sum_{i=1}^{m}\{\mathbb{I}[D_{w_1}(\widetilde{x}_i)]g_1(\widetilde{x}_i) == 1 \&\& \mathbb{I}[D_{w_2}(\overline{x}_i)]g_2(\overline{x}_i) == 1\}}{\sum_{i=1}^{m}\{\mathbb{I}[D_{w_1}(\widetilde{x}_i)]g_1(\widetilde{x}_i) == 1\}}, \tag{3}$$

$$\text{Accuracy\_}O = \frac{\sum_{i=1}^{m}\{\mathbb{I}[D_{w_1}(\widetilde{x}_i)]g_1(\widetilde{x}_i) == 0 \&\& \mathbb{I}[D_{w_2}(\overline{x}_i)]g_2(\overline{x}_i) == 0\}}{\sum_{i=1}^{m}\{\mathbb{I}[D_{w_1}(\widetilde{x}_i)]g_1(\widetilde{x}_i) == 0\}}, \tag{4}$$

$$\text{Accuracy\_}N = \frac{\sum_{i=1}^{m}\{\mathbb{I}[D_{w_1}(\widetilde{x}_i)]g_1(\widetilde{x}_i) == -1 \&\& \mathbb{I}[D_{w_2}(\overline{x}_i)]g_2(\overline{x}_i) == -1\}}{\sum_{i=1}^{m}\{\mathbb{I}[D_{w_1}(\widetilde{x}_i)]g_1(\widetilde{x}_i) == -1\}}, \tag{5}$$

$$\text{Percentage\_}H = \frac{\sum_{i=1}^{m}\{\mathbb{I}[D_{w_1}(\widetilde{x}_i)]g_1(\widetilde{x}_i) == 1\}}{m}, \tag{6}$$

$$\text{Percentage\_}O = \frac{\sum_{i=1}^{m}\{\mathbb{I}[D_{w_1}(\widetilde{x}_i)]g_1(\widetilde{x}_i) == 0\}}{m}, \tag{7}$$

$$\text{Percentage\_}N = \frac{\sum_{i=1}^{m}\{\mathbb{I}[D_{w_1}(\widetilde{x}_i)]g_1(\widetilde{x}_i) == -1\}}{m}, \tag{8}$$

in which $\{\overline{x}_i, \widetilde{x}_i\} \sim \rho_{\pi_2}$ denotes a batch of data sampled by $\pi_2$. The results are shown in Figure 2. The results depicted not only the accuracies of $\mathbb{I}[D_{w_2}]g_2$, but also the changes of these three areas during the policy learning. We can see that the accuracies in each area and the percentage of $O$ will decrease at first. While at the same time, the percentage of $H$ will increase. This is because the successful detection of $H$ will decrease the estimated percentage of $O$ and reduce the accuracy of $\mathbb{I}[D_{w_2}]g_2$. With the help of query operations, the accuracy of $\mathbb{I}[D_{w_2}]g_2$ will gradually increase. Also, followed by the learning procedure of the policy $\pi_2$, more and more $H$ will be recognized as $O$, with less and less $N$. This is why in the following period, the percentages of $H$ and $N$ will decrease while that of $O$ will increase. These results also verify that our algorithm IWRE can indeed detect $H$, $O$, and $N$ successfully as the learning process of the policy.

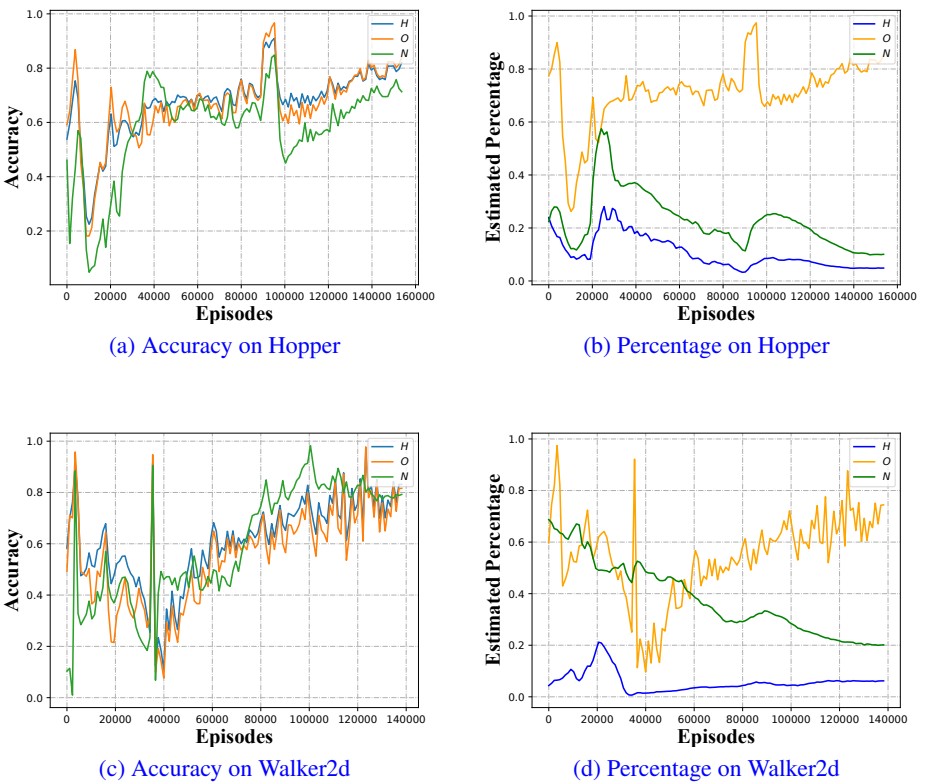

(a) Accuracy on Hopper        (b) Percentage on Hopper

(c) Accuracy on Walker2d        (d) Percentage on Walker2d

Figure 2: The accuracy and percentage of $H$, $O$, and $N$ calculated by $\mathbb{I}[D_{w_1}(\widetilde{x}_i)]g_1(\widetilde{x}_i)$ and $\mathbb{I}[D_{w_2}(\overline{x}_i)]g_2(\overline{x}_i)$ during policy learning.

## 6 Query Efficiency

We also investigate whether our query strategy is efficient. To this end, we allocate the query budget, i.e., limiting the query ratio for each method. For TPIL, it preferentially queries those data with low $D_{w_\phi}$ output; for our method IWRE, it preferentially queries those data with high $\langle D_{w_2}, g_2 \rangle$ output. Besides, since GAIL and IW cannot directly perform queries, we design a random-selection strategy for them as GAIL-Rand and IW-Rand: for each batch of data, we randomly select data and input the $\mathcal{O}_E$ observations of these data to $D_{w_1}$. If $D_{w_1}(\overline{x}) > 0.5$, which means $D_{w_1}$ regard this data being belonging to the expert demonstrations, then we would label this data as the expert data to update $D_{w_2}$. The results are depicted in Figure 3.

We can observe that the random strategy does not always improve the performance of GAIL and IW. For GAIL-Rand, without importance-weighting to calibrate the learning process of the reward function, its performance become even worse on *Hopper*, *Swimmer*, and *Walker2d*, because the queried information enhances the discrimination ability of reward function, making it even more impossible for the agent to obtain effective feedbacks; for IW-Rand, its performance is better than GAIL-Rand on most environments, and is reinforced on *Hopper*, *Reacher*, and *Walker2d*, which further demonstrate that the query operation is indeed necessary for HOIL problem, but still fails compared with our method; for TPIL, it is comparable with IW-Rand, however, its performance improvement is very limited as the budget increases, and on *Swimmer* and *Walker2d* there even exist performance degradations, which suggests that its query strategy is very unstable; for GAMA, it has a good start point, but the performance gain is very limited while the budget increases; for our method, its performance is almost the same as that of IW-Rand without query on most environments. When it is allowed to query $\mathcal{O}_E$ observation, our method outperforms other methods with a large gap, which shows that the query strategy of our method is indeed more efficient.

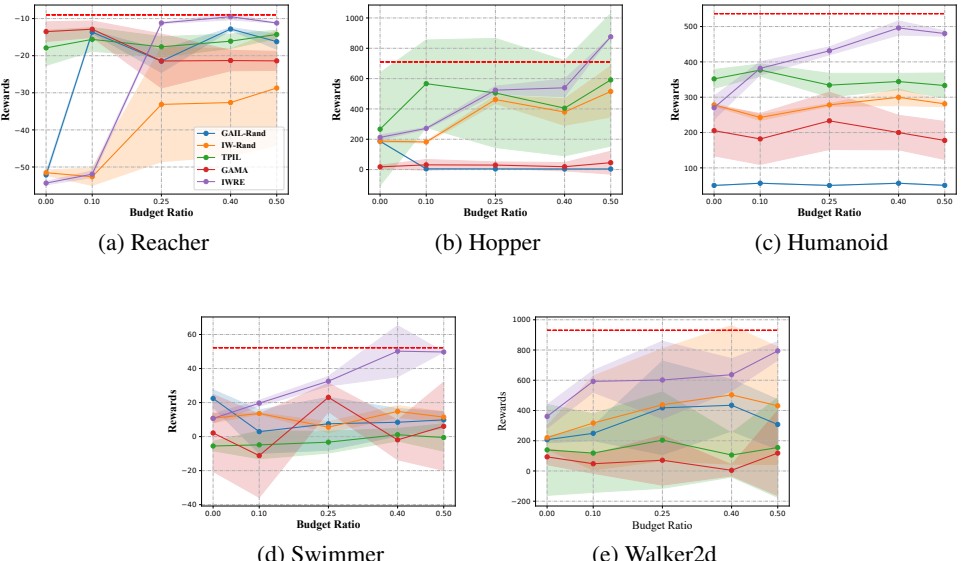

(a) Reacher        (b) Hopper        (c) Humanoid

(d) Swimmer        (e) Walker2d

Figure 3: The final rewards of each method on MuJoCo with different budget ratios, where the shaded regions indicate the standard deviation. The red horizontal dotted line represents the averaged performance of the expert.

## 7 Imitation with Different Number of Expert Trajectories

The performances of different numbers of expert trajectories of all contenders are reported in Figure 4. Each experiment is conducted 5 trials with different random seeds. We can observe that even with a very limited number of trajectories, our algorithm achieves better performance than other algorithms in most environments.

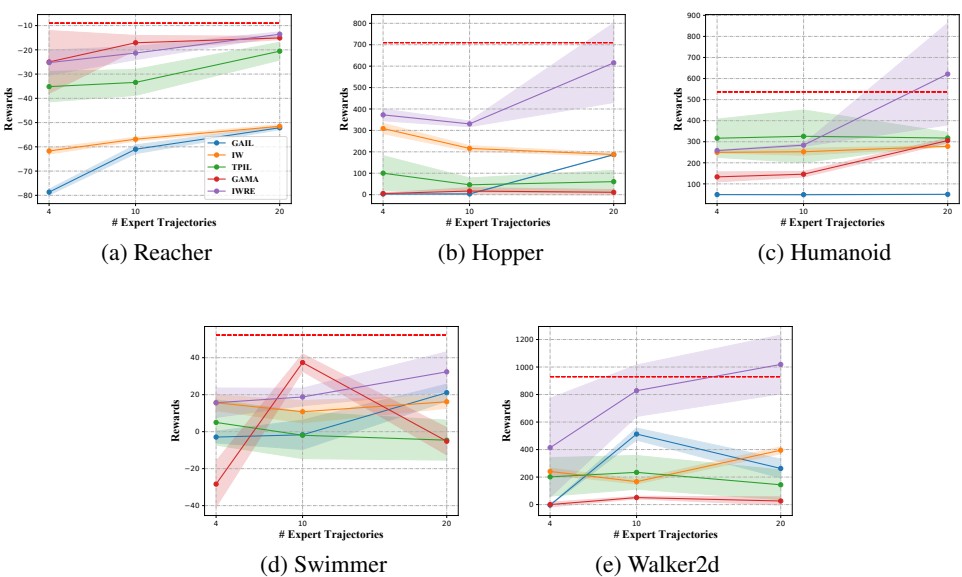

(a) Reacher        (b) Hopper        (c) Humanoid

(d) Swimmer        (e) Walker2d

Figure 4: The learning curves of each method in MuJoCo environments with different number of expert trajectories, where the shaded region indicates the standard deviation.

## 8   Notations

The notations of the main paper are gathered in Table 4.

Table 4: The notations of the main paper.

| Notation | Meaning |
| --- | --- |
| $\mathcal{S}$ | State space |
| $\mathcal{A}$ | Action space |
| $\mathcal{O}$ | Observation space |
| $\mathcal{O}_{\mathrm{E}}$ | Observation space of the expert's view |
| $\mathcal{O}_{\mathrm{L}}$ | Observation space of the learner's view |
| $\mathcal{P}$ | Transition Probability |
| $\gamma$ | Discounted factor |
| $\pi_{\mathrm{E}}$ | Expert policy under $\mathcal{O}_{\mathrm{E}}$ |
| $\pi_1$ | Auxiliary policy under $\mathcal{O}_{\mathrm{E}}$ |
| $\pi_2$ | Target policy under $\mathcal{O}_{\mathrm{L}}$ |
| $f_{\mathrm{E}}$ | Mapping function $\mathcal{S} \to \mathcal{O}_{\mathrm{E}}$ |
| $f_{\mathrm{L}}$ | Mapping function $\mathcal{S} \to \mathcal{O}_{\mathrm{L}}$ |
| $\widetilde{\mathcal{T}}_{\pi_{\mathrm{E}}}$ | Trajectory sampled by $\pi_{\mathrm{E}}$ under $\mathcal{O}_{\mathrm{E}}$ (demonstrations) |
| $\widetilde{\mathcal{T}}_{\pi_1}$ | Trajectory sampled by $\pi_1$ under $\mathcal{O}_{\mathrm{E}}$ |
| $\widetilde{\mathcal{T}}_{\pi_2}$ | Trajectory sampled by $\pi_2$ under $\mathcal{O}_{\mathrm{E}}$ |
| $\overline{\mathcal{T}}_{\pi_1}$ | Trajectory sampled by $\pi_1$ under $\mathcal{O}_{\mathrm{L}}$ |
| $\overline{\mathcal{T}}_{\pi_2}$ | Trajectory sampled by $\pi_2$ under $\mathcal{O}_{\mathrm{L}}$ |
| $x$ | An instance of state-action pair |
| $\widetilde{x}$ | An instance of observation-action pair under $\mathcal{O}_{\mathrm{E}}$ |
| $\overline{x}$ | An instance of observation-action pair under $\mathcal{O}_{\mathrm{L}}$ |
| $\rho_{\pi_{\mathrm{E}}}$ | Occupancy measure of the expert policy $\pi_{\mathrm{E}}$ |
| $\rho_{\pi_1}$ | Occupancy measure of the auxiliary policy $\pi_1$ |
| $\rho_{\pi_2}$ | Occupancy measure of the target policy $\pi_2$ |
| $D_{w_1}$ | Adversarial model on $\widetilde{\mathcal{T}}_{\pi_{\mathrm{E}}}$ and $\widetilde{\mathcal{T}}_{\pi_1}$ |
| $D_{w_2}$ | Adversarial model on $\overline{\mathcal{T}}_{\pi_1}$ and $\overline{\mathcal{T}}_{\pi_2}$ |
| $\alpha$ | Importance-weighting factor |
| $H$ | Latent demonstration |
| $O$ | Observed demonstration |
| $N$ | Non-expert data |
| $g_1$ | rejection model under $\mathcal{O}_{\mathrm{E}}$ |
| $g_2$ | rejection model under $\mathcal{O}_{\mathrm{L}}$ |

## 9   Broader Impacts

In this work, we introduce the Heterogeneously Observable Imitation Learning (HOIL) framework and propose the IWRE approach to solve the HOIL problem. Meanwhile, as collecting heterogeneous demonstrations is much more convenient than gathering homogeneous ones, this work could lead to potential risks of abusing unauthorized data. While we believe that developing these techniques is still necessary for the importance of solving imitation learning under heterogeneous observation spaces. On the other hand, there have been many techniques for preserving data privacy, which can be compatible with our approach to avoid such problems.