# OpenReview forum: "Seeing Differently, Acting Similarly: Heterogeneously Observable Imitation Learning"
_NeurIPS.cc/2022/Conference — NeurIPS 2022 Submitted_

### Official Review · Reviewer_msPH · 2022-07-08

**Rating:** 6
**Confidence:** 2
**Soundness:** 2 fair
**Presentation:** 2 fair
**Contribution:** 3 good

**Summary:**

The authors consider the problem of imitation learning, focusing on the challenging scenario where (i) the demonstrator and the learner act under different observation spaces, and (ii) there is only limited access to the demonstrator observations. The focus of the paper is thus on a paradigm the authors call 	Heterogeneously  Observable Imitation Learning: they propose methods for overcoming the challenges of the setting, and experimentally demonstrate their good behaviour on different tasks.

**Questions:**

- “…another is that in HOIL, instead of only considering the difference between two fixed policies, dealing with the dynamic change of the learner’s policy during learning is essential.”. I got the point but the sentence is unclear
- “… While in HOIL, expert’s and learner’s observations are totally different from each other, in the sense that the observations themselves are not the obstacle for acting optimally.” This is unclear to me
- AN state-action pair
- A Trajectory should be probably defined as a set T = {x_i} I=…
- X hat and h bar might become x_E and x_L?
- Why pi hat instead of pi_L?
- GAIL should be explicitly mentioned in the related works
- Dynamics mismatch and the support mismatch are often mentioned but with no introduction. A definition may be beneficial for the non-expert user
- On the experiments, some guide on how to read and interpret the plots may be useful for the understanding. Also it would be good to have a visual impression on the “category” of the other approaches (e.g. do they consider heterogenous states? Do they have access to the whole experts observations?) for an immediate comparison
- Also 20 trajectories seem to me not enough to reach conclusions on the benefit of the method. Some comments on this point are needed

**Limitations:**

There are no comments on the  potential negative societal impact of the work. It is suggested to touch the point, focusing in particular on possible application domains of interest (e.g. autonomous driving as suggested by Fig. 1).

**Strengths And Weaknesses:**

To the best of my knowledge, the work has a good level of originality on the theoretical side. Overall the structure of the manuscript is very good, with a clear storytelling. However, some parts are more clear than others. I find for instance the theoretical formulations a bit hard to follow (although I recognise that the theoretical part of the work requires it) while the experiments may be probably presented in a more interpretable way for the unfamiliar reader.
My impression is that the task considered by the authors present interesting challenges and deserves attention, and the advances with respect to existing IL approaches are convincing in Fig. 2. However, the authors should take care of the section of related works, in which the writing is not that clear.
When needed, I clarify my observations in the Questions.

---

> ### Author Response · Authors · 2022-08-02
> **Author response**
>
> Thank you for your constructive comments. Since HOIL involves two spaces, it needs detailed notations for the analysis. We have improved the clarification and added the notation tables as well as the definitions of the key issues of HOIL in the revision. Below we focus on questions other than clarification.
>
> ---
>
> **Q1: "20 trajectories seem to me not enough to reach conclusions on the benefit of the method. Some comments on this point are needed"**
>
> **A1:** 20 trajectories are commonly used in the experiments of previous imitation learning works [1, 2], So here we obeyed the same rule. Also, we provided the experiments with different numbers of expert trajectories in the supplementary materials, which drew the same conclusions.
>
> ---
>
> [1] Generative Adversarial Imitation Learning. NIPS 2016.
>
> [2] Imitation Learning from Pixel-Level Demonstrations by HashReward. AAMAS 2021.

---

> > ### Comment · Reviewer_msPH · 2022-08-06
> > **Response**
> >
> > I thank the authors for their clarifications

---

### Official Review · Reviewer_3iqV · 2022-07-11

**Rating:** 6
**Confidence:** 4
**Soundness:** 2 fair
**Presentation:** 3 good
**Contribution:** 3 good

**Summary:**

The authors propose a new challenging task for imitation learning: heterogeneously observable imitation learning. It is common that the demonstrator and the learner have totally different observation spaces, but it will make the imitator act sub-optimally because of dynamics mismatch and support mismatch. The authors propose the Importance Weighting with Rejection (IWRE) to solve the dynamics mismatch with importance sampling and support mismatch via rejection learning. They conduct experiments on Mujoco and Atari games and their method gets the best imitation performance.

**Questions:**

1. There is no explicit ablation study on your method. The effects of importance weighting and rejection learning are not clear from your current experiments.
2. The experiment design does not clearly verify the arguments in the paper. It is not clear why the method works only from the imitation performance and a simple t-SNE visualization.
3. Some visualization or analysis results to demonstrate that $I(D(x))$ and $g(x)$ correctly find the $H$, $O$, and $N$.
4. Why is there no standard deviation of IWRE and GAIL in Humanoid result (Figure6 (e))?
5. The Mujoco setup is strange. I think it is better to conduct experiments in naturally designed HOIL environments, such as BEV and first-person-view like LBC paper.


**Limitations:**

The limitation is mainly in experiment design and setup. Please see above.

**Strengths And Weaknesses:**

Strengths:

The proposed heterogeneously observable imitation learning is interesting and challenging. The proposed method is novel and performs well across many different environments.

Weaknesses:

1. The experiment setup for Mujoco is strange. I’m sure that the last half of the original observations in MuJoCo are the features of velocity. I think it is unlikely to predict the actions only from the velocity information. Would you please make more clarification about this?
2. The paper lacks sufficient analytical experiments to demonstrate their motivation and arguments. Why the method works is not clear.

Minor issues:

1. The $w_{2}$ under the $max$ should be $w_{1}$ in Eq(5) and Eq(6).
2. A ")" is missing in Eq(14), Eq(15) and Eq(16).

---

> ### Author Response · Authors · 2022-08-02
> **Author response**
>
> Thank you for your constructive comments. We would like to clarify some misunderstandings in the review below. Also, we have improved the paper and added some new experiments based on constructive suggestions. We hope the clarifications and revisions would alleviate your concerns.
>
> ---
>
> **Q1: "The experiment setup for Mujoco is strange. I’m sure that the last half of the original observations in MuJoCo are the features of velocity. I think it is unlikely to predict the actions only from the velocity information. Would you please make more clarification about this?"**
>
> **A1:** Actually we did not directly cut the original Mujoco observations into two halves under the original order. The expert and learner observations were randomly selected from the original ones. The details of selection for each environment were reported in the supplementary material. We also reported the comparable performance of RL method on each observation space to make sure the fairness of the selection.
>
> ---
>
> **Q2: "The Mujoco setup is strange. I think it is better to conduct experiments in naturally designed HOIL environments, such as BEV and first-person-view like LBC paper."**
>
> **A2:** We note that we did report experimental results for natural HOIL: the observation spaces in the Atari experiments, the visual and RAM observation spaces, are natural ones. Furthermore, as stated in A1, in Mujoco, we use the comparison of the performance of RL methods to ensure the fairness of the observation space selection. Thus we believe that the results are convincing to verify the effectiveness of IWRE.
>
> ---
>
> **Q3: "There is no explicit ablation study on your method. The effects of importance weighting and rejection learning are not clear from your current experiments."**
>
> **A3:** The explicit ablation studies are already done actually. In the experiments, "IW" is the ablation version of IWRE with importance-weighting calibration only; on the other hand, the rejection part is used to calibrate the situation of importance-weight, so it may not be realistic to just utilize the rejection part as the ablation study.
>
> ---
>
> **Q4: "The experiment design does not clearly verify the arguments in the paper. It is not clear why the method works only from the imitation performance and a simple t-SNE visualization."**
>
> **A4:** More experimental results can be found in the supplementary materials, including comparisons on query efficiency, comparisons with the different number of expert trajectories, and observation selection fairness on Mujoco. Also, we added a new experiment about visualizations of $H$, $O$, and $N$ estimation. More details can be found in A5.
>
> ---
>
> **Q5: "Some visualization or analysis results to demonstrate that $I(D(x))$ and $g(x)$ correctly find the $H$, $O$, and $N$."**
>
> **A5:** We thank the reviewer for providing such a constructive suggestion. We have added the visualization experiments to verify whether the combined model $\mathbb{I}[D_{w_2}]g_2$ can find $H$, $O$, and $N$ correctly. The results are updated in Section 5 and Figure 2 in the supplementary materials. The results depicted not only the accuracies of $\mathbb{I}[D_{w_2}]g_2$, but also the changes of these three areas during the policy learning. We hope these visualization or analysis results can alleviate your concerns.
>
> ---
>
> **Q6: "Why is there no standard deviation of IWRE and GAIL in Humanoid result (Figure6 (e))?"**
>
> **A6:** We thank the reviewer for pointing out this issue. The standard deviation shadow of IWRE in figure 6(e) was lost because of the error in PDF generation. The updated revision has fixed this issue, and the conclusion remains the same. Meanwhile, actually there exist standard deviations of GAIL at the beginning of training steps (you can find them when zooming in the figure). However, under such a complex environment of Humanoid, the learner's performance will degenerate quickly without the calibration of the importance weights.

---

> > ### Comment · Reviewer_3iqV · 2022-08-09
> > **Response**
> >
> > I appreciate the authors for the clarification and the new experiment. I think the new visualization compensates for the experiment section to verify your claims, and thus I recommend that you can put it in the main paper in your final version.

---

> > > ### Author Response · Authors · 2022-08-09
> > > **Author response**
> > >
> > > We thank Reviewer 3iqV for the positive feedback and constructive suggestions! Besides, we would like to ask if there are other concerns that make Reviewer 3iqV still feel borderline reject.

---

### Official Review · Reviewer_pD56 · 2022-07-12

**Rating:** 6
**Confidence:** 3
**Soundness:** 3 good
**Presentation:** 2 fair
**Contribution:** 2 fair

**Summary:**

This paper proposes an algorithm for Heteogeneously Observable Imitation Learning (HOIL), a variant of imitation learning in which the demonstrator and imitator have different observation spaces, and it is expensive to obtain paired samples that depict corresponding imitator and demonstrator states. Specifically, in the HOIL setting, the data available to the agent includes:

1. A small number of paired training samples (corresponding observations under the two observation spaces, along with actions) obtained from some policy $\pi_1$ (which may not be the expert policy).
2. The ability to roll out policies under the demonstrator's observation space alone.
3. The ability to query which expert observation corresponds to a given demonstrator observation.

The proposed method is a form of GAIL that uses importance sampling to reweight the demonstrator/novice observations from data source (2) so that they approximately follow the marginal distribution of the expert policy in the novice observation space. When a given imitator observation falls outside the support of the previously recorded paired data (i.e. when the importance ratio is infinite), the agent actively queries for the corresponding demonstrator observation and adds it to the available paired training data. Experiments on MuJoCo and Atari against various baseline methods for dealing with different observation spaces suggest that this approach yields higher return with fewer paired (expert/novice) observation samples.

**Questions:**

Please see the questions above. I'll copy the specific questions most important to my score here to help the authors prioritise:

- Thoroughness: on L271, how was the ratio of 1/4 chosen? What happens under different ratios?
- Thoroughness: why is PPO-RAM performing worse than IWRE?
- Significance: when would we expect sampling from the expert observation space to be higher cost than getting equivalent samples from the demonstrator observation space? When would the cost of active expert observation queries be lower than that of active expert action queries?
- Limitations: below I ask for failure cases for IWRE. It would be good to know whether the authors encountered any environments where IWRE didn't work well for reasons that are intrinsic to the approach (as opposed to, e.g., a lack of hyperparameter tuning).

**Limitations:**

The paper is explicit in setting out its assumptions, but does not do much to delineate the set of problems where IWRE can be expected to perform well. In particular, it's not clear how the environments in Figure 6 were chosen from their respective benchmark suites (Atari/Gym MuJoCo)---what happened on other environments? Are there environments where IWRE does particularly poorly? I know that it is common to only report successful environments in RL papers, but this does lead to a warped view of how effective algorithms actually are on novel problems.

I do not see any obvious ethical issues or anticipate negative social impact specific to this method.

**Strengths And Weaknesses:**

**Originality:** There is a substantial amount of existing work on imitation learning from demonstrations that are provided in a different observation space to the one available at evaluation time for the imitator. As I understand it, the novelty of this paper is in (1) explicitly treating the required number of paired observations (i.e. corresponding imitator/demonstrator observations) as something that should be minimised by the learning algorithm, and (2) proposing a combination of importance sampling and active learning to achieve this goal. The proposed approach to this problem seems quite novel to me. However, I'm not convinced of the desirability of minimising the amount of paired data, or of the assumption that active querying for paired data is cheap (this is mentioned more in "significance", below).

**Clarity:** The paper is a bit notationally heavy (here are many subscripts and accents without obvious mnemonic meanings), so it took me a while to digest the meat of the method. Here are some suggestions for things that could be clarified in the paper:

- L6/L115: the paper refers to observation spaces "coexisting". After reading the paper, I think "the two observation spaces coexist" means "the agent can sample paired observations from both observation spaces"; however, I found this terminology confusing at first, since observation spaces are just sets, and it's not clear what it means for two sets to "coexist".
- L11: "dynamics mismatch" is referred to many times but doesn't actually seem to be defined in the paper. What does this term mean?
- Figure 2: it's not clear at this point in the paper what $\pi_1$, $\pi_2$, etc. are, what the script $T$s are, or why they have bars over them. Explaining this in section 1 or moving Figure 2 to later in the paper would be helpful.
- L62-64: "dealing with the dynamic change of the learner's policy during learning is essential" --- what does this actually mean? In what sense does adversarial domain adaptation only need to consider the difference between "two fixed policies"?
- L113: "evolving data" is an odd name for ${\mathcal T}_{\pi_1}$ given that the dataset in question seems to be fixed at the beginning of training, and doesn't "evolve" in any obvious way.
- Figure 4: the customary symbol for the empty set is $\varnothing$ (`\varnothing`), not the Greek letter $\phi$ (`\phi`). The latter is currently used throughout the paper.
- Figure 4: $H$, $O$ and $N$ should be defined at the beginning of the caption, before the explanation of the figure.
- Equations (14)-(16): is $\langle \mathbb I(D^*_w), g^*\rangle$  actually an inner product between two vectors, or is it just multipyling two scalars together? I thought the latter, but the notation suggests they are higher dimensional. (same issue for all three equations)
- Definition (3): IMO it would be easier to read this as $\operatorname{supp}(\rho_{\pi_E}) \cap \operatorname{supp}(\rho_{\pi_1})$ (or at least have the equivalence made explicit).
- Eqn. (18): where are the $x_i$s sampled from?
- In general: it would be helpful to have a table of notation in the appendix.

**Thoroughness/rigour:**  After getting through the notation, I felt that the proposed method made sense. I appreciate that the method explicitly allows for active querying in situations where the support of the expert observations and novice observations do not overlap; without this, it seems like it would be theoretically impossible to solve the proposed problem with importance weighting. I also appreciated the range of expert baselines that take very different approaches to the problem of observation space mismatch (the proposed method is quite complicated, and so it's good to see that complexity justified by a comparison to a broad range of baselines).

Some concerns about thoroughness and rigour:

- L271: how was the ratio of 1/4 chosen? This choice feels somewhat arbitrary, and I feel like Figure (6) does not do a great job of communicating the tradeoff between taking samples from $\pi_1$ and $\pi_2$. Re-plotting the graphs with different ratios would be useful, or perhaps using a different metric like "number of samples of $\pi_1$ vs. number of samples of $\pi_2$ required to get to performance level $x$".
- Figure 6: why is PPO-RAM regularly performing worse than IWRE in most of these plots? It seems like that method should provide an upper bound on performance, given that it doesn't have to learn a reward function.

**Significance:** The empirical results of the paper are quite strong according to the chosen metric (return as a function of samples, with paired/expert observations being more costly). The overall significance of the paper to the field depends on how meaningful this metric is. I feel that this could be better justified in the beginning of the paper. Specifically:

- L28 claims that the observation space of the expert is "often high cost" for the learner to sample from. For what applications in this true? Are these also applications where active querying of expert observations is easy? Just having a motivating application to think about would be useful. I don't really understand the current self-driving car motivation at the beginning of the paper because it doesn't seem like sticking a camera on the dashboard or recording the instrument panel is especially expensive (it's also very hard to actively query "expert" observations in this setting).
- L124 claims that the cost of querying for expert observations is "much lower" than querying for expert actions in a given state (like DAgger does). It's not clear to me when or why this would be the case (in part because I'm not sure what would be "expensive" about reconstructing the expert observation). Again, a motivating example would be useful as an intution pump.

---

> ### Author Response · Authors · 2022-08-02
> **Author response**
>
> Thank you for your constructive comments. We have improved the clarification of our paper according to your suggestions. Below we focus on the questions about the setting and the experiment details.
>
> ---
>
>
> **Q1: "when would we expect sampling from the expert observation space to be higher cost than getting equivalent samples from the demonstrator observation space?"** and **"L28 claims that the observation space of the expert is "often high cost" for the learner to sample from. For what applications in this true? Are these also applications where active querying of expert observations is easy?"**
>
> **A1:** These assumptions are motivated from real-world problems. To see that sampling from expert observation space would cost higher than the demonstrator’s, let us take autonomous driving (AD) that we use in the introduction as an example: Existing studies pointed out that comparing to mature techniques such as using bird-eye map (BEM) and radar as the machine observations, recovering the human observations used for driving is usually more challenging and expensive [1, 2]. We have to use high-cost 3D camera systems if we want to perfectly model human perceptions, including 360-degree depth information, adverse weather, and extreme lighting conditions. It is very meaningful to avoid, at least to reduce, the usage of these systems to increase their lifespans. We also note that adopting active querying does not mean that it is easy to do this, but it is unavoidable for solving the support mismatch in HOIL, as studied in our work. So we propose the active query strategy to effectively reduce the number of queries.
>
> ---
>
> **Q2: "When would the cost of active expert observation queries be lower than that of active expert action queries?"** and **"L124 claims that the cost of querying for expert observations is "much lower" than querying for expert actions in a given state (like DAgger does). It's not clear to me when or why this would be the case (in part because I'm not sure what would be "expensive" about reconstructing the expert observation)."**
>
> **A2:** To query expert actions, we always need to query expert observations first. This is why action queries are always more expensive than observation queries. We provide the reason why reconstructing expert observation is expensive in A1. We also have improved the clarifications in this part to avoid misunderstandings.
>
> ---
>
> **Q3: "below I ask for failure cases for IWRE. It would be good to know whether the authors encountered any environments where IWRE didn't work well for reasons that are intrinsic to the approach (as opposed to, e.g., a lack of hyperparameter tuning)."**
>
> **A3:** Even though IWRE performs better than previous approaches in HOIL, there are indeed real-world situations beyond its reach. For example, if the learner’s observation space is very insufficient, then we cannot recover the expert’s policy in principle. This is beyond our assumption on IWRE such that the learner’s observation space is complete. Thus IWRE could possibly fail in such a case. We think this is a very good future research topic in the line of research for HOIL. Thanks for your good suggestion. We would include discussions as well as additional experimental results in further revisions.
>
> ---
>
> **Q4: "why is PPO-RAM performing worse than IWRE?"**
>
> **A4:** We note that PPO-RAM is directly trained under the RAM observation space, while IWRE uses demonstrations from PPO expert trained under the visual observation space. The visual PPO expert can achieve much better performance than PPO-RAM, and IWRE can effectively learn from visual demonstrations. We believe that this result is a strong verification of the meaningfulness of HOIL: it allows the expert to act under their own preferred observation spaces, while the learner can learn under a different observation space, in which the expert likely performs very bad.
>
> ---
>
>
> **Q5: "on L271, how was the ratio of 1/4 chosen? What happens under different ratios?"**
>
> **A5:** We have indeed run more ratios in many tasks. The results are not much different from those for 1/4, so we believe that 1/4 could be enough to validate the efficiency of our approach. On the other hand, since the numbers of environments, contenders, and trials are very large, we did not have enough time to run more ratios during rebuttal. Thank you for your constructive suggestion. We would include the results with more ratios in the final paper in further revisions.
>
> ---
>
> [1] A Survey on 3D Object Detection Methods for Autonomous Driving Applications. IEEE Transactions on Intelligent Transportation Systems.
>
> [2] Multi-View 3D Object Detection Network for Autonomous Driving. CVPR 2017.

---

> > ### Comment · Reviewer_pD56 · 2022-08-09
> > **Reviewer response**
> >
> > Thank you to the authors for responding to all of my questions! I'm still leaning in favor of acceptance based on the technical merits of the paper, but I'm still just as confused about the motivation to the paper as before (which I elaborate on below).
> >
> > > We have to use high-cost 3D camera systems if we want to perfectly model human perceptions, including 360-degree depth information, adverse weather, and extreme lighting conditions.
> >
> > I am confused by this comment, for two reasons:
> >
> > 1. Humans do not have 360 degree depth perception. If the goal was to see what humans see (in the literal sense of obtaining equivalent sense data to the human), then couldn't we just strap a high-quality stereo camera to the driver's head and get them to drive around? I agree that humans are better than robots at doing tasks like object recognition or driving, but humans can still do those tasks via a camera feed. Whatever the problem is in this motivating example, it is not that the human and robot have "totally different observations of the same state" (L27).
> > 2. How would you do active querying with such sensors? Wouldn't you have to drive to a particular patch of road and then turn on all the sensors at that point? Surely it would be easier to keep the sensors active at all times.
> >
> > > To query expert actions, we always need to query expert observations first
> >
> > Why? The response above claims that you need "360-degree depth" to construct expert observations, but a human can clearly give a driving demonstration without needing to fit lidar to their car (this is how driving lessons work!). Even if you did need complicated sensors to simulate the performance of the human visual system, you could still skip this step and do DAgger without needing to worry about human vision at all.
> >
> > At a high level, I think that having a more concrete example in the introduction of the paper would be helpful. What is the expert's observation space? What is the learner's observation space? Why is it expensive to obtain observations under the expert's observation space, but not under the learner's observation space? How would you efficiently do active queries for expert observations? At the very least, line 29 should explain how citations [6] and [10] support the claim that querying expert observations is expensive.
> >
> > > We have indeed run more ratios in many tasks
> >
> > Why do you need to rerun the experiments at all? I thought the 1/4 ratio was effectively just shortening or lengthening parts of the curves in figure 6, in which case the analysis could be done post-hoc for different ratios by adjusting the x-axis scale in different ways. Am I misunderstanding what is going on in these plots?

---

> > > ### Author Response · Authors · 2022-08-10
> > > **Author response**
> > >
> > > We appreciate Reviewer pD56 for the positive feedback and detailed discussions on the example of HOIL! These comments would help us improve the clarifications of our motivation. We apologize for not having clarified the differences and misunderstandings clearly enough in the original version and we see how that caused your confusion. Below we would like to answer these questions and clarify some misunderstandings.
> > >
> > > ---
> > >
> > >
> > > **Q1: " I agree that humans are better than robots at doing tasks like object recognition or driving, but humans can still do those tasks via a camera feed."**
> > >
> > > **A1:** We were pleased to learn that the reviewer agreed with us on the idea that "humans are better than robots at doing tasks like object recognition or driving". Also thanks for discussing the scenario about the generations of expert's observations, which is indeed relevant to HOIL in a broad sense. We will add these discussions in the revision. However, this scenario actually falls into a different setting from HOIL of our work. We now clarify and compare them as follows.
> > >
> > > **The setting in the Reviewer pD56 response:** Human experts use the on-board front camera feed or simulator to make decisions. The data is recorded by the camera or simulator as the demonstration data.
> > >
> > > **The setting of HOIL:** Human experts directly make decisions on a car, while some high-fidelity sensors, like 3D cameras, are used to recover human observations. The data is recorded by these high-fidelity sensors as the demonstration data.
> > >
> > > The observations of these two settings are totally different. Meanwhile, this difference can lead the totally different policies that the expert will act on, especially under complex autonomous driving tasks. Hence, we can see that these two settings are actually different (again we apologize for not having included and discussed them initially). In this sense, the observations of the expert and the learner can be totally different, like in HOIL. Also, since these sensors are expensive, we need to design the proper query strategies to reduce their use of them.
> > >
> > >
> > > ---
> > >
> > > **Q2: "How would you do active querying with such sensors?"**
> > >
> > > **A2:** Our algorithm IWRE is an online algorithm, so as the rejection part. So in real-world applications, like the autonomous driving task, we can wake these expert's view sensors and use them to gather a batch of data as soon as the model tells us to do so (e.g., $\mathbb{I}[D_{w_2}]g_2 = 1$). No need to replay the roll-out process. While the rest of the time, these sensors can be kept dormant to reduce usage costs and extend their operating lifespan.
> > >
> > > ---
> > >
> > > **Q3: "Even if you did need complicated sensors to simulate the performance of the human visual system, you could still skip this step and do DAgger without needing to worry about human vision at all."**
> > >
> > > **A3:** We describe the DAgger process as being in the training phase of HOIL, in which only the learner's observations ($\mathcal{O}_\mathrm{L}$) and not the expert's observations ($\mathcal{O}_\mathrm{E}$) are available. At the same time, algorithms like LBC require an oracle under expert observation ($\mathcal{O}_\mathrm{E}$) to provide benchmark actions for DAgger learning. This is why we stated, "To query expert actions, we always need to query expert observations first". Thank you very much for your discussions and the constructive suggestions about the concrete example. We will clarify this process in further revision to avoid potential misunderstandings.
> > >
> > > ---
> > >
> > > **Q4: "Why do you need to rerun the experiments at all? Am I misunderstanding what is going on in these plots?"**
> > >
> > > **A4:** We indeed need to rerun the whole experiment for different ratios. Each different ratio corresponds to a different auxiliary policy $\pi_1$, while a different $\pi_1$ generates completely different initial data in the pretraining phase (the original version of the data is called evolving data, and we have modified the name according to your suggestion), thus corresponding to a completely different process of HOIL. So for each ratio, we need to rerun the experiment.

---

### Author Response · Authors · 2022-08-02
**General comments**

We would like to thank all reviewers for their constructive comments and detailed feedback. We have uploaded the revision that incorporated reviewer suggestions. The updates are highlighted in blue. The major updates to the paper are as follows:

1. **Improved clarifications**.

2. **Added experiments about the estimation of $H$, $O$, and $N$ by $\mathbb{I}[D_{w_2}]g_2$**. These results can be found in Section 5 and Figure 2 in the supplementary materials.

3. **Added definitions of dynamics mismatch and support mismatch**. For now we put these definitions in Section 2 in the supplementary materials due to space limitation. We will try to add them into the main paper of the further revision if possible.

4. **Gathered the information of contenders and all notations in the supplementary material**. They can be found in Section 3 and Section 8 in the supplementary materials.

---

### Meta-Review · Area_Chair_wX7g · 2022-08-26

**Recommendation:** Reject
**Confidence:** Less certain

**Metareview:**

In this paper, the authors tackle the problem of demonstrators and learners having different observation spaces, by proposing an importance-weighted learning algorithm to bring the support of the imitator state marginal closer to that of the expert demonstrator's state marginal.

All reviewers have voted to weak accept, but it would seem that this year's NeurIPS mechanism of reviewer assignment has resulted in a much higher acceptance rate than typical, with 80%+ of the AC batch having accept votes. As such, I am tasked with the tough job of rejecting some papers that reviewers were only lukewarmly excited about. It pains me to do this to a paper that reviewers have found methodologically correct. I will be recommending this paper be rejected based on calibration against other papers I'm AC'ing, mostly for the following reason:

The HOIL paper assumes a problem setting where:

1. expert observations with more privilege observations than learner:
2. access to expert observations being high cost and invasive
3. importance-weighting the data to close the support between learner and expert state distributions mitigates (1) and (2)

However, it's not demonstrated that (1) and (2) is an actual problem in practical applications. Is sensor mismatch between human and autonomous vehicles the actual problem for learning? Do self-driving cars even utilize a policy formulation explored in the HOIL paper? How big are these state / support mismatches in practice, and couldn't they be mitigated by simpler methods? This paper is the first I've heard suggested that sensor mismatch between humans and autonomous vehicle sensors is the bottleneck for performance. If we remove this motivation from the paper and only consider the importance weighting algorithm used for removing out of distribution examples, then it doesn't feel quite as novel as there are many works that propose some kind of distribution-projection step as a way to mitigate state marginal differences between teachers and learners (e.g. CQL).

In fact, the setting (1) is not only avoided, but actively exploited by some learning papers (Asymmetric Actor Critic, Guided Policy Search, and other ideas) to *boost* the sample efficiency of learning, with the idea that the asymmetry in state support allows experts to provide useful learning signal in a way that the learner cannot adversarially overfit easily.

From reviewer pD56,
```
I didn't find the authors' response to be particularly convincing (the self-driving example doesn't seem like a good fit for what they're doing, and I think they're conflating differences in perceptual accuracy with differences in sensing hardware).
```


**Award:**

No

---

### Decision · Program_Chairs · 2022-09-14

Reject